# Operator-Based Nonlinear Control for a Miniature Flexible Actuator Using the Funnel Control Method

## Keisuke Ueno, Shuhei Kawamura and Mingcong Deng *

Department of Electrical and Electronic Engineering, Graduate School of Engineering,
Tokyo University of Agriculture and Technology, 2-24-16 Nakacho, Koganei-shi, Tokyo 184-8588, Japan;
s178312q@st.go.tuat.ac.jp (K.U.); s195704q@st.go.tuat.ac.jp (S.K.)
* Correspondence: deng@cc.tuat.ac.jp; Tel.: +81-42-388-7134

**Abstract:** Recently, the studies of soft actuators have been getting increased attention among various fields. Soft actuators are very safe for fragile objects and have an affinity to humans because they are composed of flexible materials. A miniature flexible actuator is a kind of pneumatically driven soft actuator. It has a bellowed shape and asymmetrical structure. This shape can generate a curling motion in two ways under positive and negative pressures with only one air tube. In the previous article, a control system using adaptive $\lambda$-tracking control was proposed. This control gain can become too large as time tends to infinity because the adaptive law exhibits a non-decreasing gain. To solve this problem, the funnel control method is proposed. The adaptive gain of this method not only increases but also decreases; however, the design scheme of the boundary function which is needed to decide on adaptive gain is not proposed here. In this article, an operator-based nonlinear control system's design and the design scheme of the boundary function using an observer are proposed. Then, the effectiveness of the proposed method is verified by a simulation and an experiment.

**Keywords:** soft actuator; modeling; operator theory; nonlinear control; right coprime factorization; passivity; funnel control; observer

## 1. Introduction

In the industrial fields, actuators are used to generate large amounts of force to operate heavy machinery, to perform extremely accurate positioning of semiconductor manufacturing equipment, and so on. In the coming years, they are expected to be used not only for such fields but also for medical and welfare fields. In these fields, human-friendly actuators such as power assisting robots or surgery support robots are required [1].

Soft actuators are made of lightweight and flexible materials and can perform complicated movements, such as expansion and contraction, bending, and twisting [2–4]. Some of them are driven by electricity, air pressure, heat, light, and so on. Pneumatic soft actuators expand and contract by controlling the air pressure applied inside. For example, there is the McKibben pneumatic artificial muscle [5–8] and a flexible micro-actuator (FMA) [9,10]. The McKibben pneumatic artificial muscle has characteristics similar to those of human muscles and has a large output per unit weight; therefore, it is expected to be applied to wearable robots. The FMA is made of fiber-reinforced rubber and can operate with multiple degrees of freedom, so it is possible to use it in narrow spaces that are out of reach for humans; however, these soft actuators have problems, such as limited miniaturization, high material costs, and more complicated control methods.

To solve such problems, a miniature flexible actuator has been developed [11,12]. The miniature flexible actuator is a pneumatically driven soft actuator made of a flexible material, such as silicone rubber. Compared to conventional actuators made of fiber-reinforced rubber, the actuator has advantages such as being cheaper and easier to miniaturize. Since the actuator can be bent greatly in two directions, it can accommodate fragile objects of various sizes.

The model of the actuator is, however, complicated due to the nonlinearity of silicone rubber, and a classical control theory is difficult to apply. To solve this problem, a control system has been proposed using robust right coprime factorization based on operator theory [13–16]. Operator theory [17–21] can guarantee the robust stability of the system with uncertainty. Passivity is also an important idea in control engineering [22]. With the previous method, a control system that satisfies passivity was designed, and adaptive $\lambda$-tracking, which is one of the adaptive control methods, was used [23]. Adaptive $\lambda$-tracking, however, has the problem that the gain increases as the target value changes and the gain diverges due to long-term operation.

To solve these problems, funnel control was proposed [24,25]. Funnel control is an adaptive control method that adjusts the gain from the relationship between the boundary function that is arbitrarily determined and the control error. The gain not only increases but also decreases; therefore, it solves the problem of the gain of adaptive $\lambda$-tracking increasing as the target value changes; however, the design method of the boundary function used to adjust the gain has not clearly been shown [24,25]. If the boundary function is fixed, the tip position of the actuator may not be controlled if the control deviation exceeds the upper and lower bounds of the boundary function when the desired value changes. To solve the problem, in this article, a nonlinear observer [26] is utilized to design the boundary function of funnel control. If the observer is used for the boundary function, the tip position of the actuator can be controlled because the boundary function also changes when the desired value changes. Besides, the stability of the proposed control system is guaranteed by designing the control system based on operator theory. The effectiveness of the proposed methods was verified by a simulation and an experiment.

## 2. Modeling

Section 2.1 describes the structure and bending motion of the actuator. Section 2.2 shows the method for modeling the actuator's characteristics. In Section 2.3, the method for modeling the pneumatic characteristics is introduced.

### 2.1. The Structure of the Miniature Flexible Actuator

The overall appearance and a side view of the actuator are shown in Figure 1 [11]. The shape of the actuator is a semicircular cylinder with a bellowed structure on one side and a flat surface on the other side [11]. This structure makes it possible to bend in two directions without using the fiber-reinforced rubber. In addition, the elasticity of the bellowed structure is greater than that of fiber-reinforced rubber; therefore, it can be bent more greatly than the fiber-reinforced rubber actuator [12]. Figure 2 shows the bending motion of the actuator [13]. When positive pressure is applied, the bellows side tends to expand, and the flat side does not expand easily; therefore, the actuator bends with its bellows outside (at 60 kPa in Figure 2). On the contrary, when negative pressure is applied, the bellows side is easy to contract, and the flat side is hard to contract; thus, the actuator bends with its bellows inside (at −20 kPa in Figure 2).

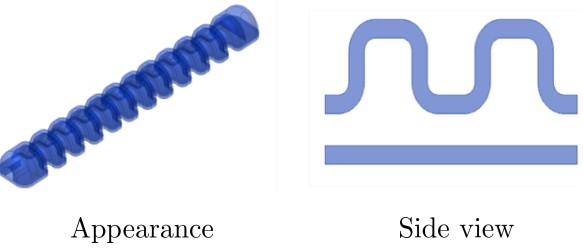

Appearance          Side view

**Figure 1.** The overall appearance and a side view of the actuator.

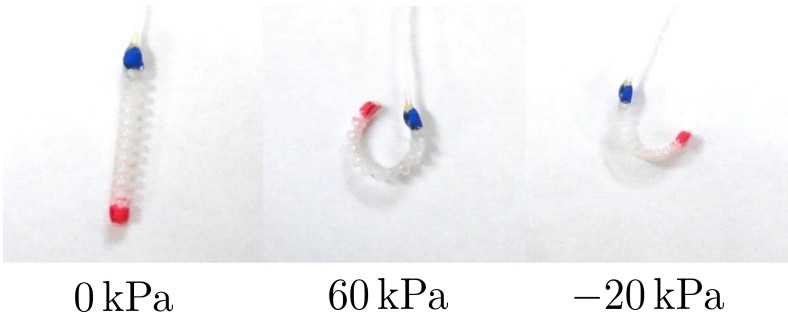

0 kPa      60 kPa      −20 kPa

**Figure 2.** The bending motion of the actuator.

### 2.2. Modeling of the Actuator Characteristics

The model of the actuator has been proposed for controlling the position of the tip (see details in [13]). As shown in Figure 3, it is supposed that the actuator curves in an arc [13]. Additionally, the bending angle $\theta \in [0, 2\pi]$, the curvature radius $R$, and the length of the actuator $L$ are defined. The origin of the $xy$ coordinate system is the tip of the actuator in the initial state. The $(x, y)$ is the coordinate of the tip of the actuator when air pressure is applied. The coordinates of the tip of the actuator at the time of deformation are determined by $R$ and $\theta$. $R$ and $\theta$ are replaced with $x$ and $y$ geometrically.

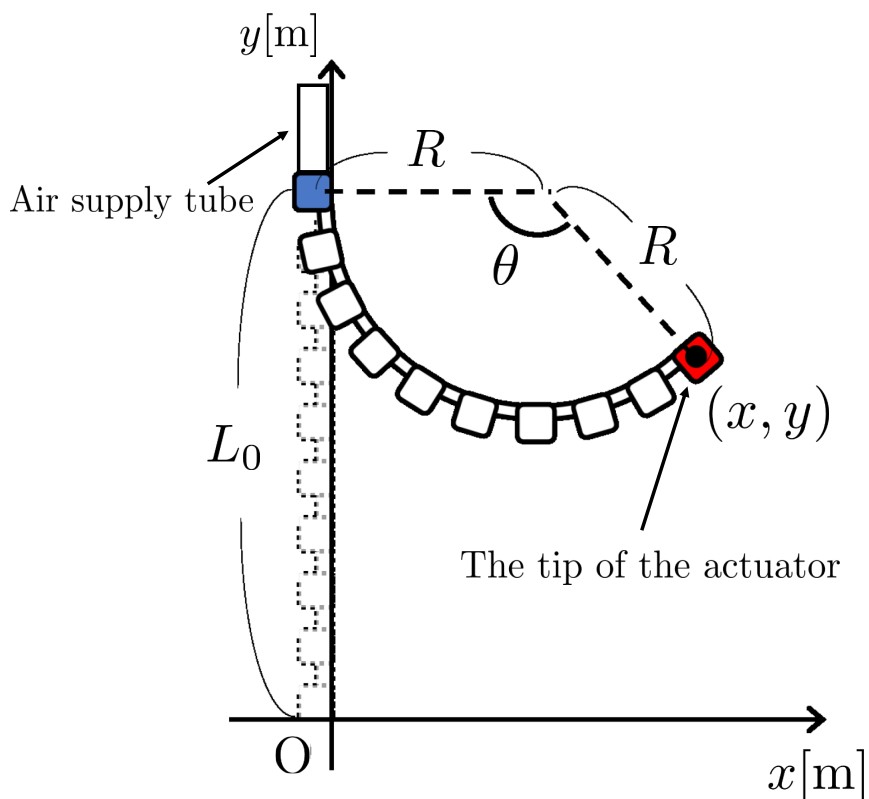

**Figure 3.** The model for analysis.

$$L = R\theta, \tag{1}$$

$$x = R - R\cos\theta, \tag{2}$$

$$y = L_0 - R\sin\theta. \tag{3}$$

The relationship between the input air pressure $p$ and the bending angle $\theta$ has been derived from the balance of moments working on bellows by applying neo-Hookean law,

which gives the strain–stress property of a single-strand of rubber [13]. The strain–stress property helps modeling to include the nonlinear elasticity of the actuator more exactly than that by Hooke's law. The relationship between $p$ and $\theta$ is shown as follows.

$$\theta = \frac{n(C_2 - \sqrt{C_2^2 - 4C_1C_3p})}{2C_1}, \tag{4}$$

where, $C_1$, $C_2$, and $C_3$ are parameters and are represented as

$$C_1 = \frac{R_2^4 - R_1^4}{2L_0^2}, \tag{5}$$

$$C_2 = \frac{3(R_2^3 - R_1^3)}{4L_0}, \tag{6}$$

$$C_3 = \frac{4\{r_2^3 - (r_1 + t_{th})^3\}}{Et_{th}}. \tag{7}$$

Table 1 shows the parameters used in the model of the actuator [13]. The detailed information about this model is written in [13].

**Table 1.** Parameters of the actuator's characteristics.

| Parameter | Definition | Value |
|:---:|:---:|:---:|
| $L_0$ | Initial length of the actuator | [m] |
| $t_{th}$ | Thickness of the rubber | [m] |
| $r_1$ | Internal radius of small chambers | [m] |
| $R_1$ | Representative radius of small chambers | [m] |
| $r_2$ | Internal radius of large chambers | [m] |
| $R_2$ | Representative radius of large chambers | [m] |
| $n$ | Number of the bellows | [-] |
| $E$ | Young's modulus | [Pa] |

*2.3. Modeling of the Pneumatic Characteristics*

In this section, the pneumatic characteristics applied to the actuator are modeled [6]. Table 2 shows the parameters used in the model of the pneumatic characteristics.

**Table 2.** Parameters of the pneumatic characteristics.

| Parameter | Definition | Value |
|:---:|:---:|:---:|
| $P$ | Air pressure of the actuator | [Pa] |
| $R$ | Gas constant | [J/Kg·K] |
| $T$ | Absolute temperature of air | [K] |
| $k$ | Heat capacity ratio of air | [-] |
| $V$ | Volume of the actuator | [m$^3$] |
| $m$ | Air flow rate | [Kg] |
| $A_0$ | Cross-sectional area of the control valve | [m$^2$] |
| $P_{tank}$ | Internal pressure of the compressor | [Pa] |
| $u$ | Input current | [mA] |
| $\beta$ | Parameter of the control valve | [Pa/mA] |
| $\gamma$ | Parameter of the control valve | [Pa] |
| $P_{max}$ | Maximum output pressure of the control valve | [Pa] |

If the compressed air is regarded as an ideal gas, the air pressure change is represented as

$$\dot{P}(t) = k_1 \frac{RT}{V(t)} m(t) - k_2 \frac{\dot{V}(t)}{V(t)} P(t),$$ (8)

where $k_1$ and $k_2 \in [1, 1.4]$ are polytropic indexes. In this article, it is assumed that the volume of the actuator does not change due to its very small size. The time change of the pneumatic input to the actuator is shown as

$$\dot{P}(t) = k_1 \frac{RT}{V} m(t).$$ (9)

In addition, the model of air flow rate is represented as

$$m(t) = \begin{cases} A_0(t) \frac{P_{tank}}{\sqrt{T}} \sqrt{\frac{k}{R} \left(\frac{2}{k+1}\right)^{\frac{k+1}{k-1}}} & \left(\frac{P_d}{P_u} \le \left(\frac{2}{k+1}\right)^{\frac{k}{k-1}}\right), \\ A_0(t) \frac{P_{tank}}{\sqrt{T}} \sqrt{\frac{2k}{R(k-1)}} \left(\frac{P_d}{P_u}\right)^{\frac{1}{k}} \sqrt{1 - \left(\frac{P_d}{P_u}\right)^{\frac{k-1}{k}}} & \left(\frac{P_d}{P_u} > \left(\frac{2}{k+1}\right)^{\frac{k}{k-1}}\right), \end{cases}$$ (10)

where $P_u$ and $P_d$ are air pressures in the upstream and downstream directions, respectively. In this article, the heat capacity ratio of air $k$ is 1.4 and $P_u$ is greater than $P_d$ because compressed air is depressurized as it passes through the control valve in the experiment system; therefore, $P_d/P_u$ satisfies the following inequality.

$$\left(\frac{P_d}{P_u} \le \left(\frac{2}{k+1}\right)^{\frac{k}{k-1}}\right).$$ (11)

The air flow rate is expressed as follows.

$$m(t) = A_0(t) \frac{P_{tank}(t)}{\sqrt{T}} \sqrt{\frac{k}{R} \left(\frac{2}{k+1}\right)^{\frac{k+1}{k-1}}}.$$ (12)

Proportional control valves are used in the pressure control system. The proportional control valve is a device that determines the degree of valve opening from the relationship between the input current and the internal air pressure. As the cross-sectional area $A_0$ of the control valve varies with the valve opening, it is represented as

$$A_0(t) = \frac{\beta u(t) - p(t) - \gamma}{P_{max}}.$$ (13)

## 3. Nonlinear Control System Using the Funnel Control Method

This section describes methods to control the actuator by using the funnel control method. Section 3.1 shows the operator-based nonlinear feedback control system's design. In Section 3.2, the passivity of the proposed system is confirmed. Section 3.3 introduces funnel control and designs a PI-funnel controller for following the target value. In Section 3.4, the design scheme of the boundary function using an observer is proposed.

### 3.1. Operator-Based Nonlinear Control Feedback System Design

Figure 4 shows the proposed nonlinear feedback control system using robust right coprime factorization based on operator theory [17,18]. The detailed information about operator theory is written in [17,18].

The given plant operator $P : U \to Y$ is said to have a correct factorization if there exist a linear space $W$ and two stable operators $D : W \to Y$ and $N : W \to Y$ such that $D$ is invertible from $U$ to $W$ and $P = ND^{-1}$ on $U$. Such a factorization of $P$ is denoted as

$(N, D)$, and the space $W$ is called a quasi-state space of $P$. In addition, $P$ is said to be a right coprime factorization, if there exist two stable operators $A : Y \to U$ and $B : U \to U$ which satisfy the Bezout identity

$$AN + BD = M \text{ for } M \in \mathcal{U}(W, U), \tag{14}$$

where $B$ is invertible, and $M \in \mathcal{U}(W, U)$ means $M$ is unimodular.

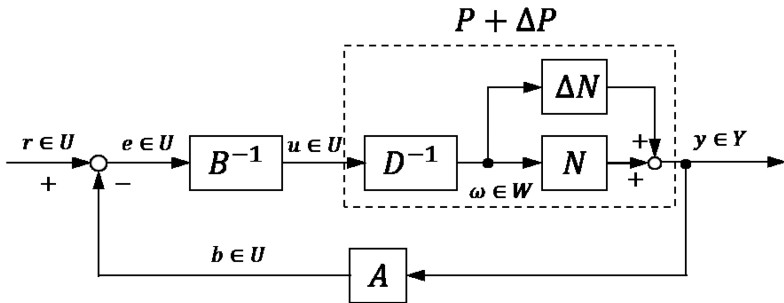

**Figure 4.** The nonlinear feedback control system.

The nominal plant $P$ is shown as

$$P : \begin{cases} \dot{x}(t) = \alpha(\beta u(t) - x(t) - \gamma), \\ y(t) = \frac{n}{2C_1}\left(C_2 - \sqrt{C_2^2 - 4C_1 C_3 x(t)}\right), \end{cases} \tag{15}$$

where the input $u = p$; the output $y = \theta$; $x$ is the state quantity; $C_1, C_2$ and $C_3$ are the same formula as Equations (5)–(7). The nominal plant $P$ is factorized into $N$ and $D^{-1}$:

$$N(\omega)(t) = \begin{cases} \dot{x}(t) = \alpha(\beta \omega(t) - x(t) - \gamma), \\ y(t) = \frac{n}{2C_1}\left(C_2 - \sqrt{C_2^2 - 4C_1 C_3 x(t)}\right), \end{cases} \tag{16}$$

$$D^{-1}(u)(t) = \omega(t) = u(t), \tag{17}$$

where $N$ is stable and $D$ is stable and invertible. $A$ and $B$ are designed to satisfy the Bezout identity in Equation (14) as follows.

$$A(y)(t) = b(t) = \frac{1}{K}N^{-1}(y)(t), \tag{18}$$

$$B^{-1}(\omega)(t) = u(t) = e(t) + \frac{\gamma}{\beta}, \tag{19}$$

where $K$ is the designed controller parameter, $A$ is stable and $B$ is stable and invertible. The actual plant has the uncertainty derived from the effect by the shape of the bellows and approximation in modeling. The plant with the uncertainty $P + \Delta P$ is shown as

$$P + \Delta P : \begin{cases} \dot{x}(t) = \alpha(\beta u(t) - x(t) - \gamma), \\ y(t) = (1 + \Delta)\frac{n}{2C_1}\left(C_2 - \sqrt{C_2^2 - 4C_1 C_3 x(t)}\right). \end{cases} \tag{20}$$

The right factorization of the nonlinear plant is shown as follows.

$$P + \Delta P = (N + \Delta N)D^{-1}. \tag{21}$$

Then, $P + \Delta P$ is factorized as follows.

$$(N + \Delta N)(\omega)(t) = \begin{cases} \dot{x}(t) = \alpha(\beta u(t) - x(t) - \gamma), \\ y(t) = (1 + \Delta)\frac{n}{2C_1}\left(C_2 - \sqrt{C_2^2 - 4C_1C_3x(t)}\right), \end{cases} \tag{22}$$

$$D^{-1}(u)(t) = \omega(t) = u(t). \tag{23}$$

When Equation (14) and

$$\|(A(N + \Delta N) - AN)I^{-1}\|_{Lip} < 1 \tag{24}$$

are satisfied, the robust stability of the plant with uncertainty $P + \Delta P$ can be guaranteed [17].

### 3.2. Passivity of the Proposed System

Passivity is important in nonlinear control, such as system stabilization and adaptive control system design [22]. The nonlinear feedback system is equivalent to the operator $NM^{-1}$ [17]. If the operator $NM^{-1}$ satisfies the passivity, the proposed system also satisfies it. The storage function $V$ of the proposed system is shown as follows.

$$V(x(t)) = \frac{1}{2}\frac{nC_3}{\alpha\beta C_2}\frac{1}{1 + \frac{1}{K}}x^2, \tag{25}$$

where $x$ is a state quantity. The differential of the storage function is represented as follows.

$$\dot{V} = \frac{nC_3}{\alpha\beta C_2}\frac{1}{1 + \frac{1}{K}}x\dot{x}, \tag{26}$$

$$= ry - \frac{\gamma C_1}{\beta nC_2}y^2 - \frac{nC_3}{\beta C_2}\left(\frac{C_2}{nC_3}y - \frac{C_1}{n^2C_3}y^2\right)^2, \tag{27}$$

$$\leq ry, \tag{28}$$

where $r$ and $y$ are the input and the output of the system. The second and the third terms in Equation (27) are quasi-negative; therefore, the operator $NM^{-1}$ is passive and the proposed system satisfies the passivity.

### 3.3. Funnel Control

Funnel control is a control method to vary the gain according to the distance between the error and the boundary function which is arbitrarily determined [24,25]. Figure 5 shows the concept of funnel control. $\overline{\mathcal{F}}(t)$ and $\underline{\mathcal{F}}(t)$ are the boundary functions and $e(t)$ is the error between the target value and the output.

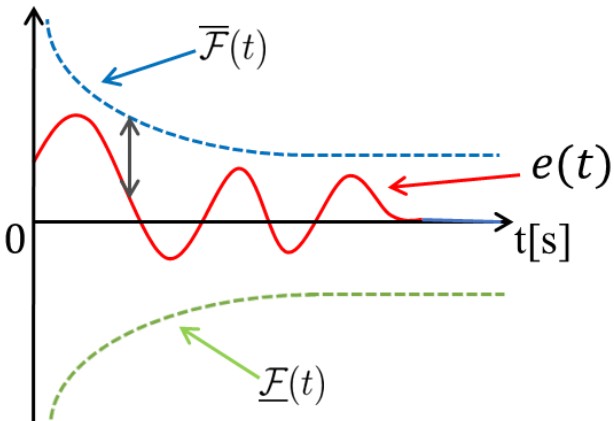

**Figure 5.** Concept of funnel control.

The adaptive gain $\kappa$ is represented as

$$\kappa(e)(t) = \begin{cases} \dfrac{\Psi(t)}{|\overline{\mathcal{F}}(t) - e(t)|} & (e(t) \geq 0), \\ \dfrac{\Psi(t)}{|\underline{\mathcal{F}}(t) - e(t)|} & (e(t) < 0), \end{cases} \tag{29}$$

where $\Psi(t)$ is the scaling function. The smaller the distance between the error and the boundary function, the larger the adaptive gain. In contrast, the larger the distance, the smaller the gain. Additionally, the error $e(0)$ at time $t = 0$ must be inside the boundary function. Therefore, the boundary function must satisfy the following.

$$\begin{cases} \overline{\mathcal{F}}(0) > e(0), \\ \underline{\mathcal{F}}(0) < e(0). \end{cases} \tag{30}$$

A funnel controller $C_f$ is shown as follows.

$$C_f(e_1)(t) = u_f = \kappa(e_1)(t)e_1(t), \tag{31}$$

where $\kappa$ is the gain and represented as Equation (29); $e_1$ and $u_f$ are the input and the output of the funnel controller, respectively. Additionally, the error remains when using only the funnel controller. To solve this problem, the funnel controller is connected to a tracking controller and extended to the PI-funnel controller $C_{PI}$ as follows.

$$C_{PI}(u_{PI})(t) = K_P u_f + K_I \int_0^t u_f dt, \tag{32}$$

where $K_P$ and $K_I$ are designed parameters; $u_{PI}$ is the output of the PI-funnel controller. Figure 6 shows a block diagram of the controller connected to a nonlinear feedback system.

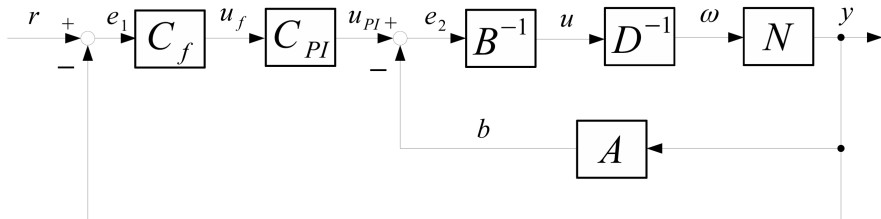

**Figure 6.** The adaptive control system.

*3.4. Design Scheme of the Boundary Function*

3.4.1. Nonlinear Observer

In this section, a nonlinear observer [26,27] used in this article is introduced. The considered nonlinear system is shown as follows.

$$\dot{x}(t) = f(x(t), u(t)), \tag{33}$$
$$y(t) = h(x(t)), \tag{34}$$

where $x(t), u(t)$, and $y(t)$ are the state quantity, the input, and the output, respectively; $f$ and $h$ are continuously differentiable in $(x, u)$. In general, the nonlinear observer is represented as follows.

$$\dot{\hat{x}}(t) = f(\hat{x}(t), u(t)) + p(y(t) - \hat{y}(t)), \tag{35}$$
$$\hat{y}(t) = h(\hat{x}(t)), \tag{36}$$

where $\hat{x}(t)$ is the estimated state quantity of the observer, $\hat{y}(t)$ is the output of the observer, and $p(y(t) - \hat{y}(t))$ is a correction function based on the error information $y(t) - \hat{y}(t)$. A performance function of the observer is defined as follows.

$$E(\hat{y}(t); y(t)) = \frac{1}{2}(y(t) - \hat{y}(t))^2. \tag{37}$$

The gradient of the performance function is calculated and the estimated state quantity of the observer is modified to reduce the performance function. The nonlinear observer using the performance function is represented as follows.

$$\dot{\hat{x}}(t) = f(\hat{x}(t), u(t)) - \mathcal{L}\nabla_{\hat{x}(t)}E(\hat{y}(t); y(t)), \tag{38}$$
$$\hat{y}(t) = h(\hat{x}(t)), \tag{39}$$

where $\nabla_{\hat{x}(t)}E(\hat{y}(t); y(t))$ is the gradient of $E(\hat{y}(t); y(t))$ with respect to $\hat{x}(t)$ and $\mathcal{L}$ is a proportional coefficient. The nonlinear observer in Equations (38) and (39) is concretely shown as

$$\dot{\hat{x}}(t) = f(\hat{x}(t), u(t)) - \mathcal{L}\frac{\partial h(\hat{x}(t))}{\partial \hat{x}(t)}(y(t) - \hat{y}(t)), \tag{40}$$

$$\hat{y}(t) = h(\hat{x}(t)). \tag{41}$$

The estimated quasi state of the observer $\hat{x}(t)$ is shown as

$$\hat{x}(t) = x(t) + e(t), \tag{42}$$

where $e(t)$ denotes the error of the estimated state quantity of the observer [27]. Taylor expansion (linear term plus second or higher order) for Equation (42) is represented as

$$\dot{\hat{x}}(t) = \dot{x}(t) + \dot{e}(t), \tag{43}$$
$$= f(x, u) + \left(\frac{\partial f(x, u)}{\partial \hat{x}(t)} - \mathcal{L}\frac{\partial h(x)}{\partial \hat{x}(t)} \cdot \frac{\partial h(x)}{\partial \hat{x}(t)}\right)e. \tag{44}$$

The differential of the estimation error $\dot{e}(t)$ is shown as

$$\dot{e}(t) = \dot{\hat{x}}(t) - \dot{x}(t), \tag{45}$$
$$= \left(\frac{\partial f(x, u)}{\partial \hat{x}(t)} - \mathcal{L}\frac{\partial h(x)}{\partial \hat{x}(t)} \cdot \frac{\partial h(x)}{\partial \hat{x}(t)}\right)e. \tag{46}$$

The Lyapunov function is defined as follows [27].

$$V = \frac{1}{2}e^2. \tag{47}$$

The differential of the Lyapunov function is shown as

$$\dot{V} = e\dot{e}, \tag{48}$$
$$= \left(\frac{\partial f(x, u)}{\partial \hat{x}(t)} - \mathcal{L}\frac{\partial h(x)}{\partial \hat{x}(t)} \cdot \frac{\partial h(x)}{\partial \hat{x}(t)}\right)e^2. \tag{49}$$

If $\mathcal{L}$ which satisfies $\frac{\partial f(x,u)}{\partial \hat{x}(t)} - \mathcal{L}\frac{\partial h(x)}{\partial \hat{x}(t)} \cdot \frac{\partial h(x)}{\partial \hat{x}(t)} < 0$ is chosen, the estimation error $e(t)$ is uniformly asymptotically stable. The detailed information about the nonlinear observer is written in [26,27].

### 3.4.2. Boundary Function Using the Nonlinear Observers

The nonlinear observers used for the boundary function are shown as follows [26,27].

$$\overline{\varnothing}: \begin{cases} \dot{\overline{x}}(t) = f(\overline{x}(t), u(t)) + \overline{\mathcal{L}} \frac{\partial(h(\overline{x}(t)) + \overline{\Delta}h(\overline{x}(t)))}{\partial \overline{x}(t)}(y(t) - \overline{y}(t)), \\ \overline{y}(t) = h(\overline{x}(t)) + \overline{\Delta}h(\overline{x}(t)), \end{cases} \tag{50}$$

$$\underline{\varnothing}: \begin{cases} \dot{\underline{x}}(t) = f(\underline{x}(t), u(t)) + \underline{\mathcal{L}} \frac{\partial(h(\underline{x}(t)) + \underline{\Delta}h(\underline{x}(t)))}{\partial \underline{x}(t)}\left(y(t) - \underline{y}(t)\right), \\ \underline{y}(t) = h(\underline{x}(t)) + \underline{\Delta}h(\underline{x}(t)), \end{cases} \tag{51}$$

where $f$ and $h$ are the plant model, $\mathcal{L}$ is the observer gain, and $\Delta h$ is the error given to the system. The boundary function is designed using these two observers. Specifically, the error between the output estimated by each observer, and the target value is utilized as the boundary function. The boundary function is represented as follows.

$$\mathcal{F}: \begin{cases} \overline{\mathcal{F}}(t) = r(t) - \underline{y}(t), \\ \underline{\mathcal{F}}(t) = r(t) - \overline{y}(t). \end{cases} \tag{52}$$

$f, h$, and $\Delta h$ in Equations (50) and (51) are shown as

$$f(x(t), u(t)) = \alpha(\beta u(t) - x(t) - \gamma), \tag{53}$$

$$h(x) = \frac{n}{2C_1}\left(C_2 - \sqrt{C_2^2 - 4C_1 C_3 x(t)}\right), \tag{54}$$

$$\Delta h(x) = \Delta \frac{n}{2C_1}\left(C_2 - \sqrt{C_2^2 - 4C_1 C_3 x(t)}\right) + \delta, \tag{55}$$

where $\Delta$ and $\delta$ are designed parameters. If the observer gain $\overline{\mathcal{L}}$ and $\underline{\mathcal{L}}$ satisfy the following, the estimation error $e(t)$ represented by Equations (50) and (51) is asymptotically stable.

$$\frac{\partial f(x, u)}{\partial \overline{x}(t)} - \overline{\mathcal{L}}\left(\frac{\partial(h(x(t)) + \overline{\Delta}h(x(t)))}{\partial \overline{x}(t)}\right)^2 < 0, \tag{56}$$

$$\frac{\partial f(x, u)}{\partial \underline{x}(t)} - \underline{\mathcal{L}}\left(\frac{\partial(h(x(t)) + \underline{\Delta}h(x(t)))}{\partial \underline{x}(t)}\right)^2 < 0, \tag{57}$$

where $\overline{\Delta}$ and $\underline{\Delta}$ are designed parameters. From Equations (53)–(57), the range of the observer gain is represented as follows.

$$-\frac{\alpha(C_2^2 - 4C_1 C_3 x_{max})}{(1 + \overline{\Delta})^2 n^2 C_3^2} < \overline{\mathcal{L}}, \tag{58}$$

$$-\frac{\alpha(C_2^2 - 4C_1 C_3 x_{max})}{(1 + \underline{\Delta})^2 n^2 C_3^2} < \underline{\mathcal{L}}. \tag{59}$$

If $\overline{\mathcal{L}}$ and $\underline{\mathcal{L}}$ which satisfy Equations (58) and (59) are selected, the estimated state quantity of the observer equals that of the plant; therefore, the output of the observer is shown as

$$\begin{aligned} \overline{y}(t) &= h(\overline{x}(t)) + \overline{\Delta}h(\overline{x}(t)), \\ &= h(x(t)) + \overline{\Delta}h(x(t)), \end{aligned} \tag{60}$$

$$\begin{aligned} \underline{y}(t) &= h(\underline{x}(t)) + \underline{\Delta}h(\underline{x}(t)), \\ &= h(x(t)) + \underline{\Delta}h(x(t)). \end{aligned} \tag{61}$$

From Equations (52), (60), and (61), the distance between the boundary function and the error is represented as follows and the gain does not diverge.

$$|\overline{\mathcal{F}}(t) - e(t)| = |r(t) - (h(x(t)) + \underline{\Delta}h(x(t))) - (r(t) - h(x(t)))|,$$
$$= |\Delta h(x(t))|, \tag{62}$$
$$|\underline{\mathcal{F}}(t) - e(t)| = |r(t) - (h(x(t)) + \overline{\Delta}h(x(t))) - (r(t) - h(x(t)))|,$$
$$= |\overline{\Delta}h(x(t))|. \tag{63}$$

Figure 7 shows the proposed system using the observer. Each operator is designed as follows.

$$F = \alpha x(t), \tag{64}$$
$$G = \alpha\beta\omega(t) - \alpha\gamma, \tag{65}$$
$$H = \frac{n}{2C_1}\left(C_2 - \sqrt{C_2^2 - 4C_1C_3x(t)}\right), \tag{66}$$
$$H + \overline{\Delta}H = (1 + \overline{\Delta})\frac{n}{2C_1}\left(C_2 - \sqrt{C_2^2 - 4C_1C_3x(t)}\right) + \overline{\delta}, \tag{67}$$
$$H + \underline{\Delta}H = (1 + \underline{\Delta})\frac{n}{2C_1}\left(C_2 - \sqrt{C_2^2 - 4C_1C_3x(t)}\right) + \underline{\delta}, \tag{68}$$

where $\overline{\delta}$ and $\underline{\delta}$ are designed parameters.

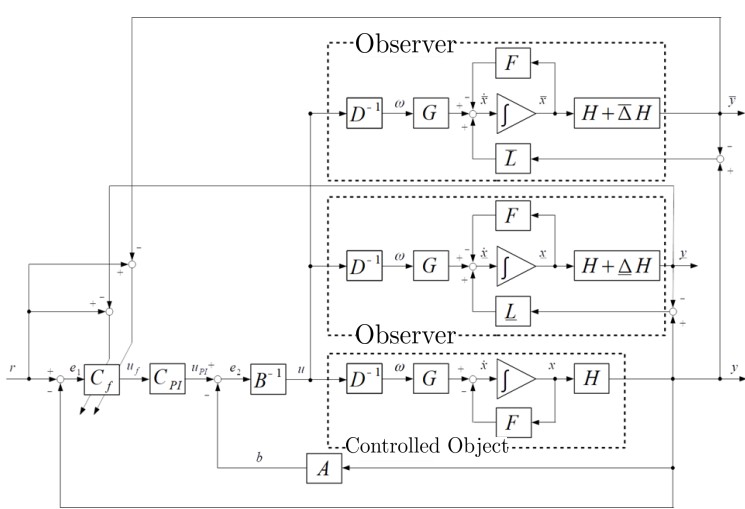

**Figure 7.** The adaptive control system using the observer.

## 4. Results and Discussion

This section shows and discusses the simulation results and the experimental results to verify the effectiveness of the proposed control system. The simulation results were obtained using MATLAB(R2017a), which is one of the most effective software products for system engineering. Section 4.1 introduces an experimental system for controlling the actuator. Sections 4.2–4.4 show the simulation and experimental results of the proposed method respectively.

### 4.1. Experimental System

Figure 8 shows the experimental system and Figure 9 shows the experimental flow [13,14]. The experimental system consisted of the actuator, an air compressor (DPP-AYAD, Koganei, Tokyo, Japan), a safety regulator (RP1000-8-07, CKD, Aichi, Japan), an electro-pneumatic regulator (ITV0010-0CS, SMC, Tokyo, Japan) providing air pressure for the actuator, a camera(HD Pro Webcam C920r, Logicool, Tokyo, Japan) measuring

the output of the actuator, and a computer sending an electrical signal. The bending angle was measured by acquiring and processing images with a camera. The tip of the actuator was colored red and the base was colored blue. Then, the red region and the blue region were extracted from the image taken by a camera, and the bending angle was calculated from the relationship between the two regions. The following explains how to move the actuator.

1.  The air compressor provides air pressure for the safety regulator.
2.  The air pressure is regulated by the safety regulator for the sake of not breaking the actuator.
3.  The computer sends an electrical signal to the electro-pneumatic regulator and decides on the opening of the valve.
4.  The air pressure is sent into the actuator and it moves.
5.  The output is captured as an image by a camera and fed back to the computer.

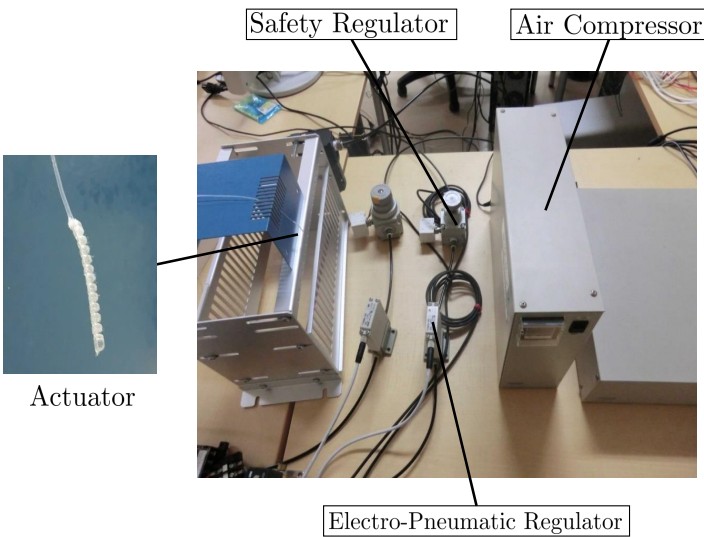

**Figure 8.** Experimental system.

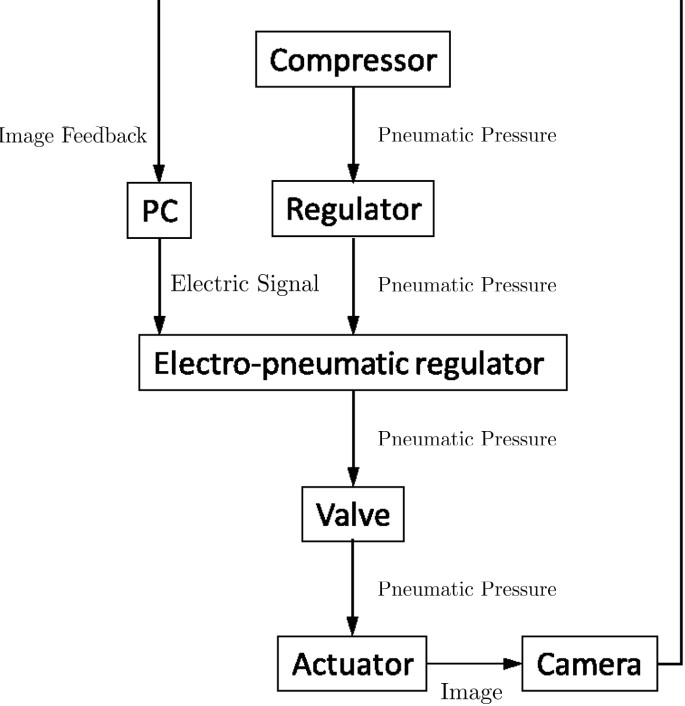

**Figure 9.** Experimental flow.

### 4.2. Parameters Used in the Simulation and Experiment

Table 3 shows the simulation and experimental parameters.

**Table 3.** Parameters used in the simulation and the experiment.

| Parameter | Definition | Value |
|:---:|:---:|:---:|
| $L_0$ | Initial length of the actuator | $0.6 \times 10^{-3}$ m |
| $t_{th}$ | Thickness of the rubber | $0.15 \times 10^{-3}$ m |
| $r_1$ | Internal radius of small chambers | $0.25 \times 10^{-3}$ m |
| $R_1$ | Representative radius of small chambers | $0.325 \times 10^{-3}$ m |
| $r_2$ | Internal radius of large chambers | $0.85 \times 10^{-3}$ m |
| $R_2$ | Representative radius of large chambers | $0.925 \times 10^{-3}$ m |
| $n$ | Number of the bellows | 12 |
| $E$ | Young's modulus | $0.95 \times 10^6$ Pa |
| $\alpha$ | Parameter of the control valve | 0.34 |
| $\beta$ | Parameter of the control valve | 6.25 Pa/mA |
| $\gamma$ | Parameter of the control valve | 25 Pa |
| $K$ | Control parameter | 40 |
| $K_P$ | Proportional parameter | 0.19 |
| $K_I$ | Integral parameter | 0.19 |
| $\overline{\mathcal{L}}$ | Designed parameter | 1 |
| $\underline{\mathcal{L}}$ | Designed parameter | 1 |
| $\overline{\Delta}$ | Designed parameter | 0.05 |
| $\underline{\Delta}$ | Designed parameter | 0.1 |
| $\overline{\delta}$ | Designed parameter | 0.3 |
| $\underline{\delta}$ | Designed parameter | $-0.3$ |

In the simulation and experiment, the scaling function of the Funnel controller was a constant multiple of the boundary function and is represented as follows.

$$\Psi(t) = 0.42\mathcal{F}(t). \tag{69}$$

### 4.3. Simulation Results

Figures 10–19 show the simulation results. Figure 11 shows the result of the proposed method and confirms that the output angle of the actuator follows the target value. Figures 12 and 13 show that the error is inside the boundary function. As shown in Figure 14, the proposed method can follow the output angle to the target value faster than the previous method [23]. From Figures 15 and 16, the gain increases whenever the target value changes in the previous method using adaptive $\lambda$-tracking control, whereas the gain increases and decreases and remains constant after the actuator follows the target value in the proposed method. Figure 17 shows that the derivative of the storage function is less than the supply rate and confirms that the proposed system satisfies the passivity. To compare the proposed method with a conventional boundary function which is fixed, Figures 18 and 19 show the results of the method without using the observers. Figure 18 shows that the output angle does not follow the desired value when the value changes. Figure 19 shows that the error deviates from the boundary function. These results show the effectiveness of the proposed design scheme.

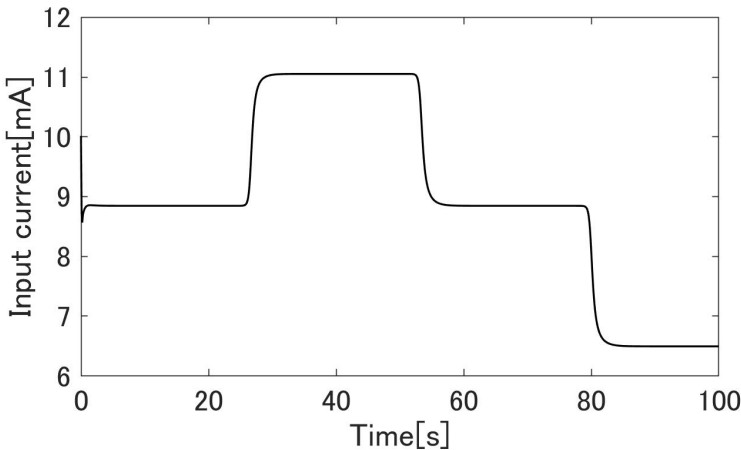

**Figure 10.** Input current $u$.

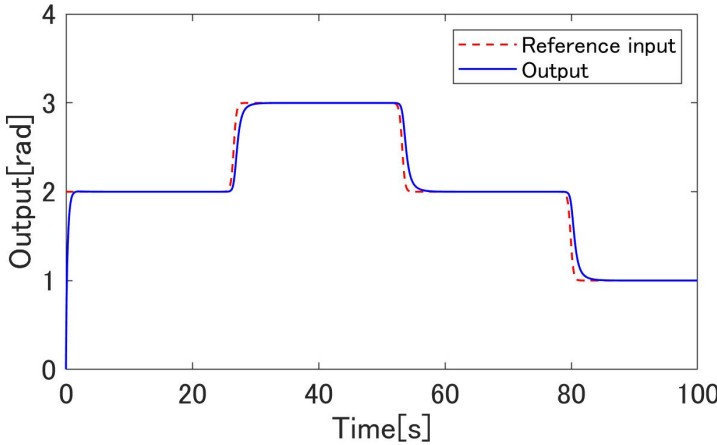

**Figure 11.** The output angle $\theta$ of the proposed method.

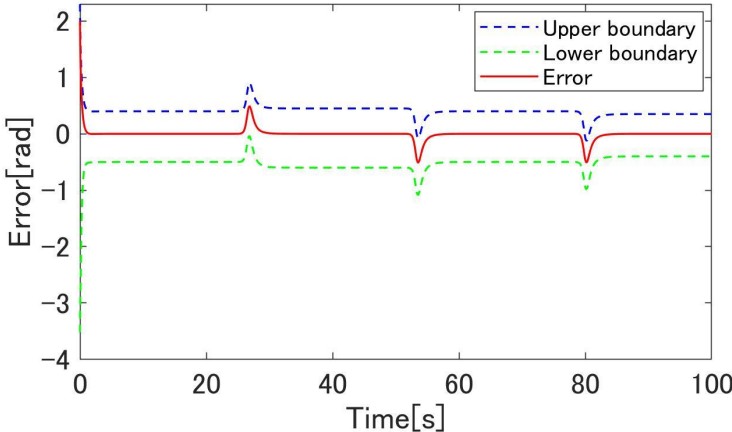

**Figure 12.** The error $e$ and boundary functions $\overline{\mathcal{F}}$ and $\underline{\mathcal{F}}$ of the proposed method.

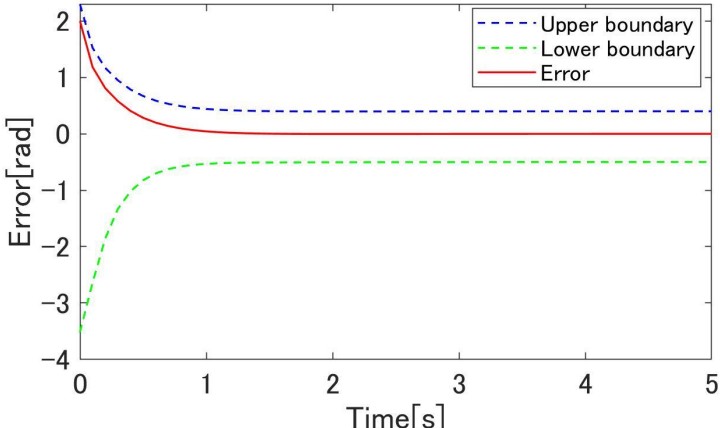

**Figure 13.** Enlarged view of Figure 12.

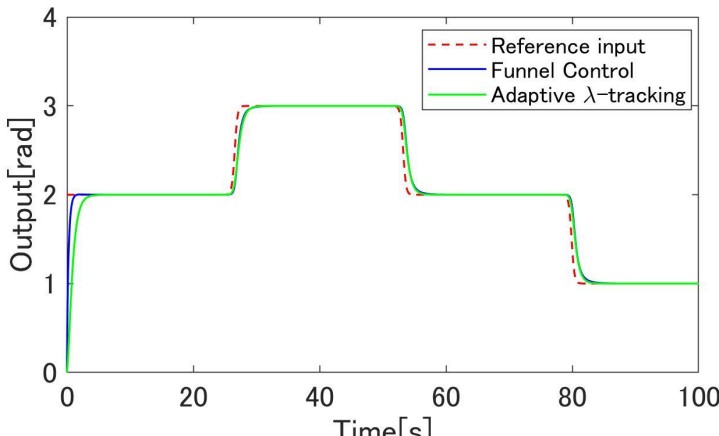

**Figure 14.** The comparison of output angles $\theta$.

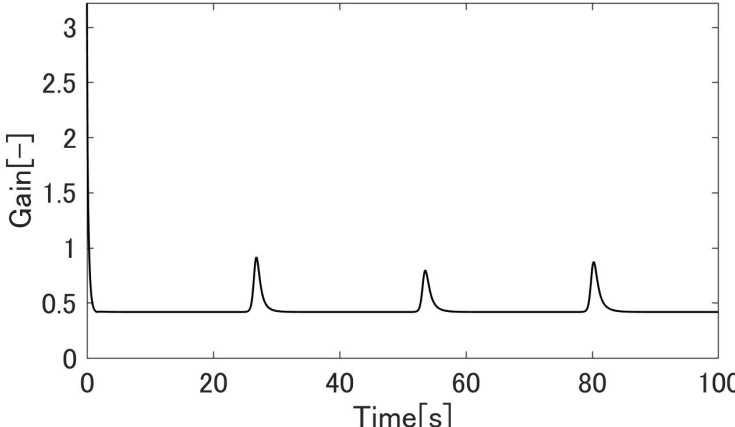

**Figure 15.** The gain $\kappa$ of the proposed method.

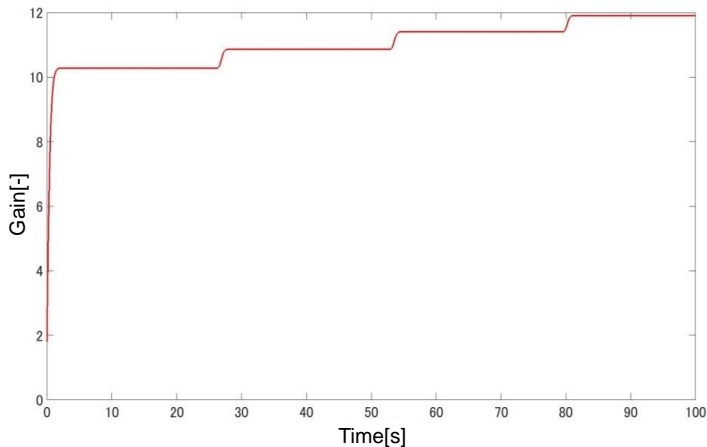

**Figure 16.** The gain of the previous method.

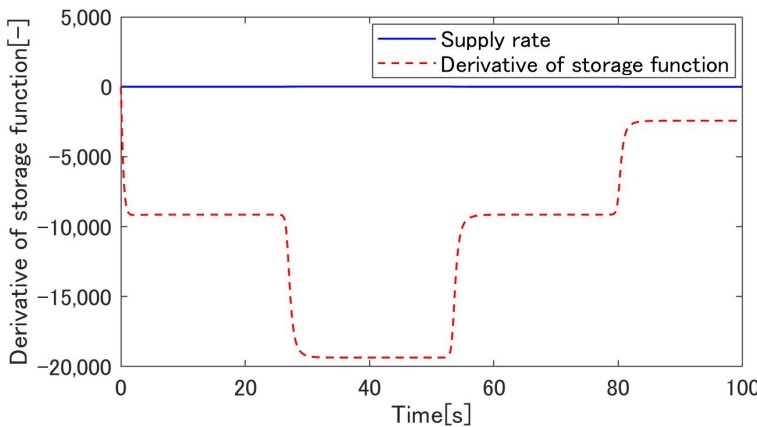

**Figure 17.** The derivative of the storage function $V$.

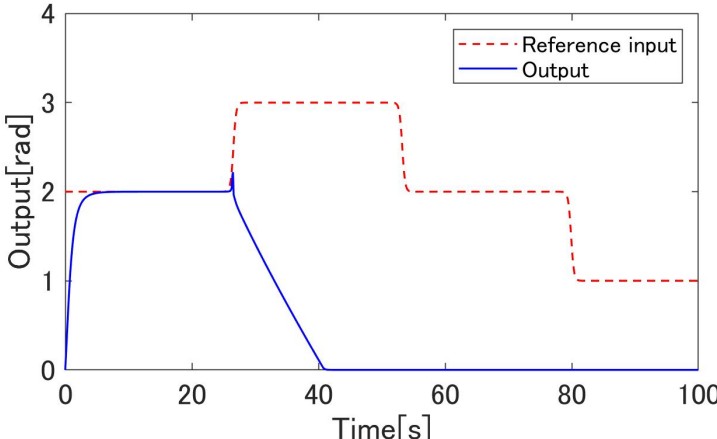

**Figure 18.** The output angle $\theta$ of the method without using the observers.

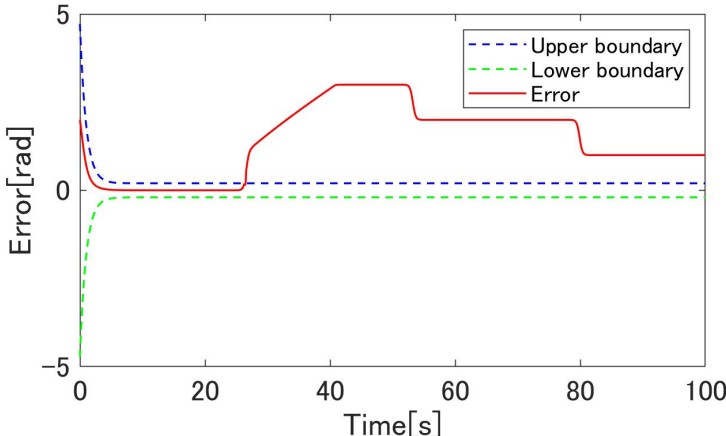

**Figure 19.** The error *e* and boundary functions without using the observers.

### 4.4. Experimental Results

Figures 20–28 show the experimental results. Figure 20 shows the result of the proposed method and confirms that the output angle follows the target value. From Figures 22 and 23, the controlled deviation is inside the boundary function. Figure 24 shows that the proposed method can follow the output angle to the target value faster than the previous method [23]. Additionally, the proposed method has a smaller steady-state error than the previous method. Figure 25 shows that the gain increases whenever the target value changes in the previous method using adaptive $\lambda$-tracking control. In contrast, Figure 26 shows that the gain increases and decreases and remains constant after the actuator follows the target value in the proposed method. Figure 27 shows that the derivative of the storage function is less than the supply rate and confirms that the proposed system is passive. Figure 28 shows that the proposed system satisfies the robust stability, as shown in Equation (24). These results show the effectiveness of the proposed method.

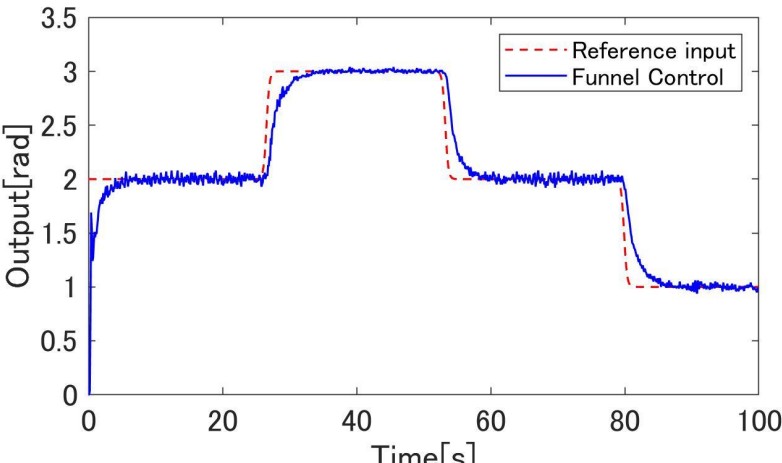

**Figure 20.** The output angle $\theta$ of the proposed method.

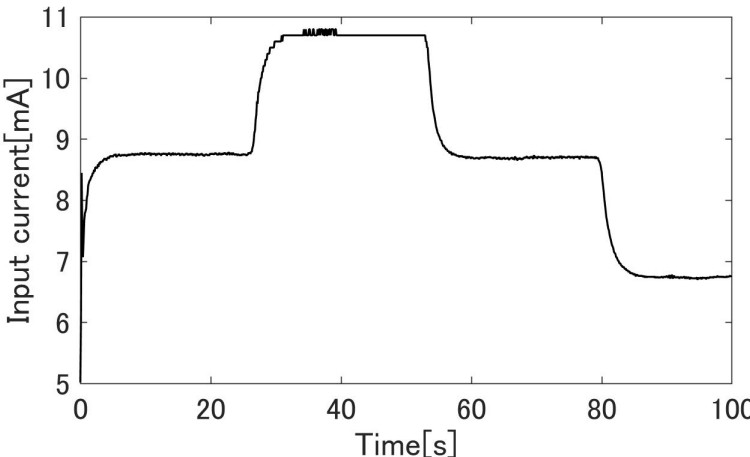

**Figure 21.** Input current *u*.

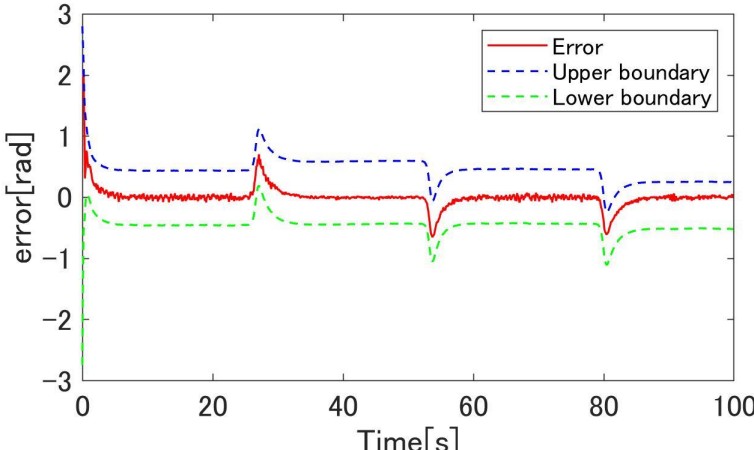

**Figure 22.** The error *e* and boundary functions $\overline{\mathcal{F}}$ and $\underline{\mathcal{F}}$ of the proposed method.

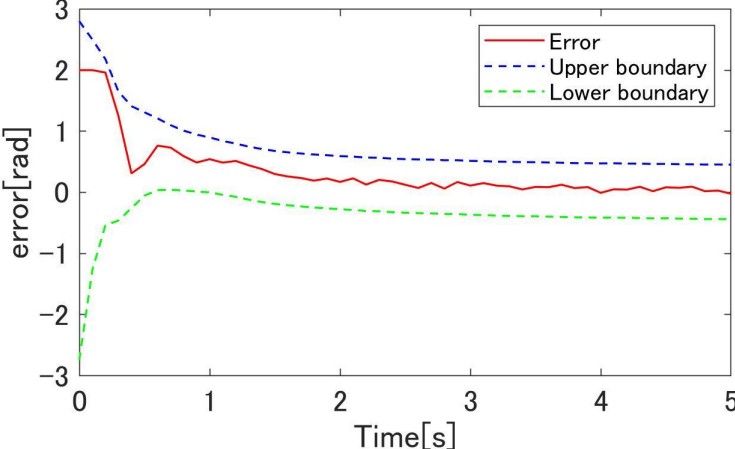

**Figure 23.** Enlarged view of Figure 22.

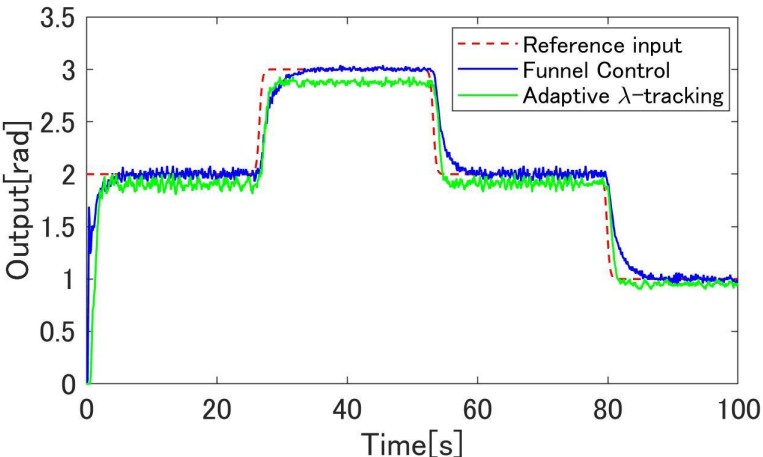

**Figure 24.** The comparison of output angles $\theta$.

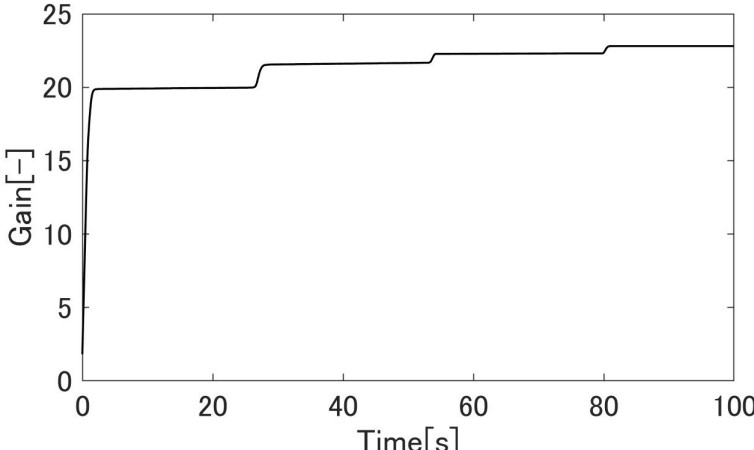

**Figure 25.** The gain of the previous method.

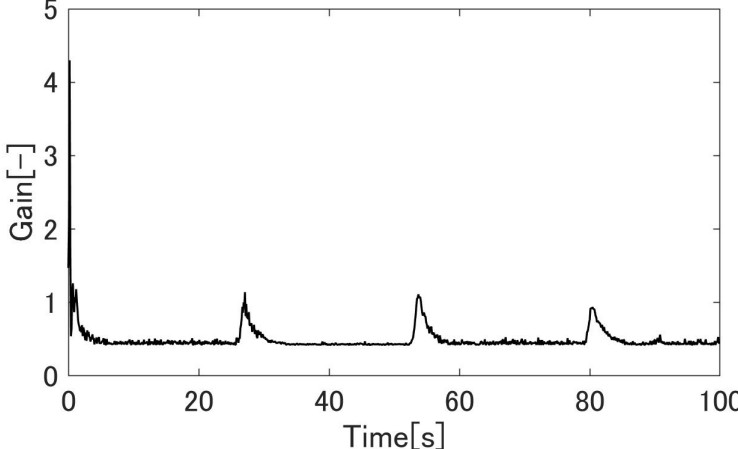

**Figure 26.** The gain $\kappa$ of the proposed method.

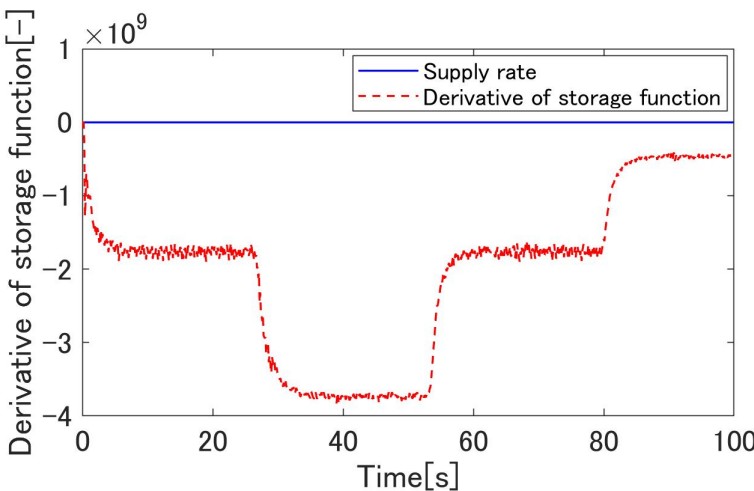

**Figure 27.** The derivative of the storage function $V$.

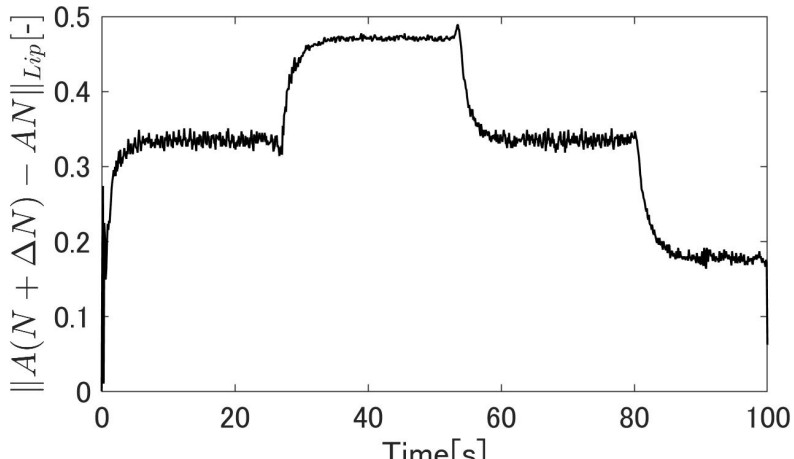

**Figure 28.** The robust stability analysis.

## 5. Conclusions

In this paper, the nonlinear control system for a miniature flexible actuator using the funnel control method is proposed. The nonlinear control system is designed using robust right coprime factorization based on operator theory, and robust stability of the system is guaranteed because the system satisfies the robust stability condition. The controller using the funnel control method solved the problem that the gain may diverge—from the previous method. Additionally, it is proposed to use an observer to design the boundary function of funnel control. In conclusion, the simulation and experimental results showed the effectiveness of the proposed method.

**Author Contributions:** K.U. proposed the nonlinear control system using the funnel control method for the actuator; S.K. wrote this paper; M.D. suggested technical support and gave overall guidance on the paper. All authors have read and agreed to the published version of the manuscript.

**Funding:** This research received no external funding.

**Institutional Review Board Statement:** Not applicable.

**Informed Consent Statement:** Not applicable.

**Data Availability Statement:** Data is contained within the article.

**Conflicts of Interest:** The authors declare no conflict of interest.

## Abbreviations

The following abbreviations are used in this manuscript:

FMA　　Flexible micro-actuator

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
