# Peer review of "Operator-Based Nonlinear Control for a Miniature Flexible Actuator Using the Funnel Control Method"

_machines, doi:10.3390/machines9020026_

Round 1

Reviewer 1 Report

The authors present an interesting control strategy to overcome increasing gain of adaptive controller for soft actuators. This paper seams to be a second part fo [23]. Also, part of the present paper is similar to [15], where a same scenario is presented.

Comments:

1) Present the variable when there are used. For example, P_{tank} is mentioned in (10), but defined few lines after, confusing the reading. The same with e_1(t). Consider modifying the script to get a fluent reading.

2) Be careful using end points. For example, before eq. (17) the sentence ends with point, when one spect no ending. Please check it. In general, review the notation (end point on equation (18) when sentence is not ended...).

3) Section 2.4 is hard to follow. What is the meaning of equation (15)?

4) Reference [17]: Nonlinear

5) Why the simulation does not consider the real value of the parameters? It can help to compare simulation and experiment. 

6) Be careful using 'the previous method', better use reference [23], it's confusing. Also, clarify the gain function and storage function. In fact, figure 17 represents the derivative, as said. Please, review the figure captions.

7) Figure 24 increase, not decrease, as say in line 138. It seems that in lines 136-138 the figure numbers are misreferencied.

8) Conclusions: robust stability is proved... where? Please clarify this point.

Author Response

For reviewer 1. Thanks.

Reviewer 2 Report

The paper is well and clearly written and the considered topic is up to date. The scientific contribution is based on a proper implementation of known , however not trivial, control techniques for a relatively new type of actuator.

My main comments are as follows.

1. In think that contents of the paper could be modified to better address the main contribution. I believe that Section 2 should be focused on a plant modeling only. Next subsections 2.4-2.7 should be a part of Section 3 which could be devoted to control/estimation problems. Then Section 4 should contain subsection 2.8 as well as results of simulations and experiments.

2. I think that it would be valuable to make a comment what kind of stability is achieved. In Eqs. (32) and (33) it can be seen that derivative of the output function with respect to estimates are taken into account. Does it mean that the observers are well defined only locally? What is a basin of attraction?

3. I also suggest to add symbols in captions of Figs 10-26 to recall meaning of signals/parameters introduced in Sections 2.2-2.7.

4. The sentence above Eq. (44): “doesn’t diverge” -> “does not diverge”

Author Response

For reviewer 2. Thanks.

Reviewer 3 Report

Flexible Actuator using Funnel Control Method” presents a study where adaptive nonlinear control strategies are used to control the position of a miniature flexible actuator.

In the reviewer opinion, the paper suffers from several drawbacks that prevent its publication. Firstly, the English level is not suitable for a journal publication as there are several sentences which are not clear or are imprecise. A non-exhaustive list of examples is presented below:

  1. “The adaptive gain of this method has not only increase gain structure but also decrease one”; - this sentence is not clear;
  2. “Actuators have mainly used motors or hydraulics…”- this sentence is not clear;
  3. “Actuators have mainly used motors or hydraulics to generate a large amount of force to operate heavy machinery or to perform extremely accurate positioning of semiconductor manufacturing equipment; however, the actuators are difficult to work with humans and handle fragile objects, therefore soft actuators that can perform tasks that were not possible are attracting attention...” – this sentence implies that soft actuators can be an alternative to handle a large amount of force or to operate heavy machinery, which does not seem to be the case in the majority of soft actuators.
  4. “The gain does not diverge because it not only increases but also decreases” The fact that an adaptive gain not only increases but also decreases does not imply it does not diverge.
  5. “however, the design method of the boundary function used to adjust the gain is not clearly shown” – shown where?
  6. “In this article, the air flow rate is assumed to be a sonic flow and expressed as follows…” – No explanation is given as to why is this assumption valid.

Several other examples exist, so the reviewer considers that the level of English is not acceptable for a journal publication.

Regarding the content, the reviewer considers that the authors failed to explain the added value of the proposed control strategy. This opinion is based on two arguments. First, although the authors compare the proposed control strategy with one previously published (and in this comparison the proposed control strategy appears to offer better results), a comparison with a datum controller (for example, a conventional PID based controller) should be made in order to assess the actual added value of the proposed control strategy, namely when compared with much simpler control strategies. Second, apparently the added value of the control strategy proposed in this work lies in the design scheme of boundary function, as the use of PI Funnel control is not new in literature. However, apparently no comparison is made with the use of more conventional boundary functions. Furthermore, the reasoning behind the use of the proposed design scheme is not clearly presented. For instance, why is the boundary function designed using the two observers presented in equations (32) and (33), i.e., what led the authors to choose such a strategy instead of others?

Given the above reasoning, the reviewer considers that the paper should not be accepted in its present form.

Author Response

For reviewer 3. Thanks.

Round 2

Reviewer 1 Report

The authors had improved the paper.